# A Comprehensive Review on the Use of Herbal Dietary Supplements in the USA, Reasons for Their Use, and Review of Potential Hepatotoxicity

Adnan Khan [1,*], Kashyap Chauhan [1], Heather Ross [2], Natalia Salinas Parra [2], John Magagna [2], Makala Wang [2], Patrick Zhu [2], Ryan Erwin [1] and Dina Halegoua-DeMarzio [3]

1 Department of Medicine, Thomas Jefferson University Hospital, Philadelphia, PA 19107, USA; kashyap.chauhan@jefferson.edu (K.C.); ryan.erwin@jefferson.edu (R.E.)
2 Sidney Kimmel Medical College at Thomas Jefferson University Hospital, Philadelphia, PA 19107, USA; heather.ross@students.jefferson.edu (H.R.); natalia.salinas-parra@students.jefferson.edu (N.S.P.); john.magagna@students.jefferson.edu (J.M.); makala.wang@students.jefferson.edu (M.W.); patrick.zhu@students.jefferson.edu (P.Z.)
3 Division of Gastroenterology and Hepatology, Department of Medicine, Thomas Jefferson University Hospital, Philadelphia, PA 19107, USA; dina.halegoua-demarzio@jefferson.edu
* Correspondence: adnan.khan@jefferson.edu

**Abstract:** Herbal and dietary supplement (HDS) use has grown exponentially in the United States. Unfortunately, the incidence of HDS-related liver injury has proportionally increased. Despite the potential for certain HDSs to cause clinically significant liver injury, they are not regulated by the Food and Drug Administration. Recent efforts have been made to regulate HDSs but are far removed from the scrutiny of prescription medications. Scant literature exists on HDSs and their risks of causing liver injury. In this comprehensive review, we examine trends of HDS use in the United States and the pathophysiologic mechanisms of drug-induced liver injury (DILI) of certain HDSs. Finally, we review usage rates; benefits, if any; purported pathophysiology of DILI; and propensity for progression to fulminant hepatic failure of nine HDSs linked to clinically significant DILI.

**Keywords:** herbal supplements; dietary supplements; chronic liver disease; end-stage liver disease; hepatotoxicity; Herbalife; Hydroxycut; drug-induced liver injury

## 1. Introduction

Consumer interest in herbal dietary supplements (HDSs) as alternative forms of or supplements to conventional medication has increased over the last three decades. The number of HDSs marketed in the United States (US) increased from around 4000 in 1993 to 55,000 in 2012 and continues to grow [1]. According to the CDC, in 2017–2018, 57.6% of adults 20 years old and over reported using HDSs in the past 30 days. HDS use increased with age, as did simultaneous use of multiple HDSs. One quarter of adults 60 years old or older reported taking four or more HDSs; women within that age range were the most likely to use HDSs, at 80.2%. From 2007 to 2018, HDS use increased in all age groups among US adults [2].

Although already on the rise in recent years, HDSs have become even more popular after emerging as potential adjunct therapies for COVID-19. A systematic review of 12 randomized control trials found that the consumption of HDSs and zinc sulfate may significantly benefit COVID-19 recovery, although these studies also recognized the low quality of the included trials [3]. These results were inconsistent with other previously published reviews [3].

Despite a lack of evidence that HDSs benefit COVID-19, consumers were drawn to these products to enhance their immune systems. After vitamin and mineral supplements, turmeric, elderberry, garlic, and echinacea were the top ingredients used for immunity [4].

Over 43% of HDS users increased their supplement regimen during the pandemic and were more likely to practice healthy lifestyle habits than their non-supplement-using counterparts [4,5]. Supplement users who were male, aged 18–34, had children in the house, or who were seasonal users altered their routine during COVID-19 more than other groups [4]. HDSs were used as both preventative and therapeutic treatments by consumers. Although there has not yet been an equivalent study in the US, a cross-sectional study of Saudi Arabian patients' HDS use before and during COVID-19 infection found a significant increase in patients using black seed and ginger [6].

Consumer interest was so strong during the pandemic that HDS sales in the US surpassed USD 10 billion for the first time in 2020, totaling roughly USD 11.261 billion, a 17.3% increase from 2019 [7]. Equally, both US retail channels, mainstream and natural, showed an increased profit margin. Mainstream channels are defined as mass market channels such as convenience and grocery stores. Natural channels are defined as natural health food specialty retail outlets. Consumers cited overall health, immune health, and cardiovascular health as their top reasons for HDS use [7]. Elderberry, apple cider vinegar, and ashwagandha sales increased most from the year prior, and elderberry, horehound, and cranberry were the top-selling HDSs of mainstream multi-outlet channels in the US [7]. The high epidemiological burden of stress, anxiety disorders, and sleep disorders due to COVID-19 likely popularized the stress-reducing and sleep-inducing effects of HDSs [8]. The rise in sales suggests increased acceptance and favorable views of HDSs by the American public, who were willing to embrace alternative treatments given the unprecedented nature of the pandemic.

## 2. Regulation of HDS

Under the Dietary Supplement Health and Education Act of 1994 (DSHEA), the Food and Drug Administration (FDA) is not required to review or approve HDSs for safety or efficacy before they are marketed [9]. Individual HDS manufacturers are only required to notify the FDA before marketing their products if the HDS contains a new dietary ingredient, defined as any dietary ingredient marketed after 1994 [9]. Otherwise, individual HDS manufacturers are responsible for determining the safety, dietary ingredient composition, and appropriate serving size of HDSs without any oversights [9]. Individual HDS manufacturers are also allowed to market HDSs with non-specific claims and benefits, such as weight loss and liver support, without documented evidence [10].

Federal law requires individual HDS manufacturers to label HDSs as food supplements and provide a full list of dietary ingredients [9]. Despite this requirement, however, many HDSs have been recalled for containing contaminated ingredients or dietary ingredients that are different from those listed on the product label [11]. Quality analysis studies have demonstrated the contamination of HDSs with various heavy metals and pesticides [12]. A recent study investigated the accuracy of dietary ingredient labels on HDSs and found a mislabeling rate of 51% [13]. All of the mislabeled HDSs in this study lacked one or more of the compounds listed on the dietary ingredient label, and some also contained one or more compounds not listed on the dietary ingredient label [13]. Mislabeling rates were highest among HDSs marketed as weight loss supplements and those with appearance- and performance-related claims [13].

The implications of HDS mislabeling and contamination include adverse events such as hepatotoxicity. Cases of HDS-induced liver injury after the consumption of yellow turmeric (Curcuma longa) supplement were identified in Scandinavian countries [14]. Further investigations identified the compound Nimesulide, a non-steroidal anti-inflammatory, present in the HDSs as the precipitating cause of drug-induced liver injury (DILI) in these cases [14]. The incidence of HDS-induced liver injury is increasing and is associated with the rise in HDS use. Of the DILI cases in the US, 20% are related to HDS-induced liver injury, and many of these cases are related to HDSs containing more than one dietary ingredient [10]. The HDSs that are the most associated with DILI include those marketed with appearance- and performance-related claims [13].

Before HDS consumption, the FDA recommends that consumers reach out to their healthcare providers to discuss safe use [11]. HDSs often contain more than one dietary ingredient that can result in unintended interactions and adverse events. HDSs can also impact the pharmacokinetics of prescription drugs, resulting in unintended HDS–drug interactions [11]. Under the Dietary Supplement and Nonprescription Drug Consumer Protection Act of 2006, individual HDS manufacturers are responsible for reporting serious adverse events that can be attributed to HDS consumption to the FDA [15]. Many of the serious adverse events reported to the FDA occurred in HDSs containing multiple dietary ingredients [16].

## 3. Hepatotoxic Effects

The liver is involved in many bodily processes, including endocrine and exocrine functions, immunologic functions, and a wide range of metabolic functions, such as the breakdown of carbohydrates, fats, and proteins. One of the liver's exocrine functions involves bile production, a mixture of bile acids, water, phospholipids, cholesterol, and electrolytes. Many drug metabolites rely on bile as a route of excretion. The liver also aids the production of albumin; clotting factors; amino acids; and urea, made from ammonia, which the liver transaminases and deaminates from amino acids [17].

Many molecules in the bloodstream are eliminated by the liver's two-phase biotransformation and elimination system. Phase I, known as hydroxylation, includes various Cytochrome P450 enzymes that add reactive oxygen sites to these target molecules. Phase II involves various conjugation enzymes that add water-soluble groups to aid in elimination. Foods and nutrients have been shown to be necessary for and able to modulate this system. Phase 1 cytochrome enzymes transform their starting molecules via oxidation, peroxidation, and reduction reactions by adding hydroxyl, carboxyl, and/or amino acid groups to their starting molecules. These reactions create electrophilic molecules that have the potential to injure their surrounding structures. Cytochrome phenotypes and numbers vary between different people, meaning that their ability to handle certain drugs and nutrients via this pathway may differ significantly. This variation can affect drug efficacy, side effect profiles, and toxicity [18].

Phase 1 enzymes: CYP1 enzymes are involved in processing hormones, various pharmaceuticals, procarcinogens, polycyclic aromatic hydrocarbons, and other environmental toxins. CYP2A-E enzymes are involved in processing ketones, glycerol, fatty acids, drugs, xenobiotics, and hormones and have been shown to have several phenotypes that affect drug metabolism. CYP3A enzymes are involved in processing caffeine, testosterone, progesterone, androstenedione, and polyaromatic hydrocarbons. CYP4 enzymes are primarily found outside of the liver and play a smaller role in drug processing. The phase 1 enzymes and their substrates are outlined in Table 1 below [18].

Phase 2 enzymes: UDP-glucuronosyltransferases are essential in processing steroid hormones and bilirubin. These enzymes covalently link glucuronic acid to their target molecule. Sulfotransferases transfer a sulfuryl group to their target molecule, which often reduces their target molecule's reactivity and toxicity. Glutathione S-Transferases are enzyme complexes that transfer a glutathione group to their target molecules and are induced when reactive oxygen species are being created. Amino acid transferases transfer an amino acid to their target molecule to aid in elimination. N-acetyl transferases transfer acetyl groups to their target molecules. Methyltransferases methylate their target molecules and play a vital role in estrogen detoxification. The phase 2 enzymes and their substrates are outlined in Table 2 below.

Other Pathways: Nrf2 is a transcription factor that helps to regulate detoxification and antioxidant systems. When activated, Nrf2 dissociates from the cytosolic protein and translocates into the nucleus to promote phase II detoxification enzymes. NrF2 is protective against stress-related conditions and toxicity from drugs or herbals. Metallothionein is also a key regulator with the ability to regulate heavy metal detoxification. The cysteine rich protein is regulated by certain stimuli such as heavy metals, oxidative stress, and zinc. In

certain conditions, such as inflammatory bowel disease, metallothionein has shown to be decreased in intestinal mucosa [18].

Mechanisms for liver injury resulting from HDSs can come in several forms. Anabolic steroids, which are sometimes marketed as "bodybuilding supplements," can cause injury to the liver via a cholestatic mechanism, leading to increased bilirubin levels and, in some cases, jaundice and itching [10]. Green tea extract (GTE), along with many other HDSs, has an unknown liver injury mechanism but causes a pattern of transaminitis similar to that of the liver injury caused by acute hepatitis [10]. Liver biopsies in patients with acute liver injury related to GTE use may show eosinophils, inflammation, and necrosis in a pattern similar to the one induced by acute hepatitis [19]. Multi-ingredient nutritional supplements (MINS) often include several ingredients, and toxicity can be caused by a specific component of the regimen or an unknown adulterant, making it difficult to determine the mechanism of injury [10].

Recent data have suggested that liver injury is immune mediated by haptenization or via molecularly mimicry and that certain drugs' reactive metabolites may stress hepatocytes to stimulate the adaptive immune system by binding to proteins, which further stimulate T-Cells to express Fas ligand (FASL) and tumor necrosis factor (TNF) to induce cell death [20,21].

**Table 1.** Cytochrome P450 enzymes that add reactive oxygen sites to these target molecules, creating more electrophilic compounds.

| Phase 1 Enzymes | |
| --- | --- |
| **CYP Family** | **Typical Substrates** |
| CYP1 | Hormones, various pharmaceuticals, procarcinogens, polycyclic aromatic hydrocarbons, and other environmental toxins. |
| CYP2A–E | Ketones, glycerol, fatty acids, drugs, xenobiotics, and hormones. |
| CYP3A | Caffeine, testosterone, progesterone, androstenedione, and polyaromatic hydrocarbons. |

Information cited from Hodges, R.E.; Minich, D.M. (2015) [18].

**Table 2.** Conjugation enzymes that add water-soluble groups to aid in elimination.

| Phase 2 Enzymes | |
| --- | --- |
| **Phase 2 Enzyme Class** | **Group Phase 2 Enzyme Adds to Target** |
| UDP-glucuronosyltransferases | Glucuronic acid |
| Sulfotransferases | Sulfuryl |
| Glutathione S-Transferases | Glutathione |
| N-acetyl transferases | Acetyl |
| Methyltransferases | Methyl |

Information cited from Hodges, R.E.; Minich, D.M (2015) [18].

## 4. Genetic and Demographic Susceptibility

The number of patients who suffer from DILI caused by HDSs is difficult to determine, as it is often unclear if patients are using these substances. However, there are data estimating that HDSs are responsible for 20–40% of acute liver failure (ALF) cases due to DILI. A Chinese study of 26,000 cases of DILI showed that only 1% of cases associated with HDSs led to ALF, but of the patients who died from ALF, HDSs were the leading cause [22]. The authors of this review investigated the demographics of patients with DILI caused by HDSs in Europe, Asia, and the US. They concluded that the typical clinical patient presentation was a young, otherwise healthy female presenting with elevated bilirubin and aminotransferases [22]. DILI in patients using HDSs for bodybuilding usually presents with increased bilirubin and prolonged jaundice in young men. Hepatocellular DILI leads

to higher rates of death and transplantation and predominantly presents in middle-aged women [23].

Differences in cytochrome P450 genetic polymorphisms affect drug metabolism and have been linked to the accumulation of toxic drug or metabolite levels, suggesting a genetic predisposition to developing hepatotoxicity [24].

Multi-ingredient products are increasingly implicated as causes of acute liver injury; green tea is a common ingredient in these products [25]. The US Drug-Induced Liver Injury Network (DILIN) performed a formal causality assessment on over 1400 patients who suffered from liver injury. Patients with green tea liver injury were often jaundiced and symptomatic and had elevated levels of serum aminotransferase with a hepatocellular pattern of injury [23]. They also had mildly elevated alkaline phosphatase levels. In patients with green tea-associated liver injury, HLA typing showed HLA-B*35:01 in 72% of cases. This type was only found in 15% of liver injury cases due to other supplements and in 12% of cases where the injury was the result of other drugs. These differences were statistically significant [25].

## 5. Pharmacokinetic and Pharmacodynamics Changes in Chronic Liver Disease

Liver disease can affect pharmacokinetics in many ways. Liver blood flow and enzyme activity are both crucial to metabolize drugs. Some drugs with a high first-past metabolism are affected by changes in liver blood flow, while drugs with "low hepatic extraction" are more affected by hepatic failure [24]. Plasma proteins are also important to pharmacokinetics and can be affected by liver function [26]. Liver impairment also has the potential to affect drug availability, distribution, and biliary elimination [27].

Patients with chronic liver disease (CLD) may have a slower metabolism in the liver, as their ability to metabolize proteins depends on their current hepatic reserve. Patients with CLD are also more likely to have low serum albumin and elevated serum ammonia levels and suffer from zinc deficiency at a higher rate, which decreases the capacity for ornithine transcarbamylase to metabolize ammonia [28]. As liver disease progresses to cirrhosis, the liver loses more function. Loss of liver parenchyma decreases the number of enzymes that are active in drug metabolism. In patients with cirrhosis, cytochrome alterations can occur. These changes are variable with different comorbid conditions and the state of current disease progression. However, there is a general pattern in patients with cirrhosis that CYP1A and CYP31 concentrations decrease while levels of CYP2C, CYP2A, and CYP2B remain at their baseline concentrations. Alcohol and aldehyde dehydrogenase levels may also be affected. Sulphation rates may decrease as well, but glucuronidation reactions are mostly unaffected by cirrhosis [29].

Studies have shown that various nutrients and HDSs, including cruciferous vegetables, resveratrol, green tea, black tea, curcumin, turmeric, soybean, garlic, fish oil, rosemary, astaxanthin, and chicory root, are inducers of CYP1 enzymes [18]. CYP1 enzyme inhibition has been found in black soybean, black tea, turmeric, apiaceous vegetables, quercetin, daidzein, grapefruit, peppermint, dandelion, kale garlic, and chamomile [18]. Inducers of CYP2 enzymes may include chicory root, quercetin, rosemary, garlic, and fish oil. Their inhibitors may include ellagic acid, green tea, turmeric, black tea, resveratrol, myricetin, watercress, chrysin, and medium-chain triglycerides [18]. Nutrients that induce CYP3 enzymes may include garlic, fish oil, rooibos tea, and turmeric. Inhibitors may include green tea, black tea, quercetin, grapefruit, myricetin, kale, and soybean. Inducers of CYP4 enzymes may include green tea and the caffeic acid found in coffee [18].

## 6. Interaction with Other Drugs

There are concerns regarding pharmacokinetic interactions between HDSs and conventional medication with the same absorption, distribution, metabolism, or excretion mechanisms. CYP1A2, CYP2C9, CYP2C19, CYP2D6, CYP2E1, CYP3A4, OATP1A1, OATP1A2, OATP2B1, and P-gp are the most well categorized to date. The six CYP enzymes account for the metabolism of approximately 80% of all prescribed drugs (Zanger UM, Schwab

M). In terms of drug metabolism, cytochrome P450 enzymes regulate gene expression and enzyme activities and impact genetic variation [30]. Two common HDSs that affect drug metabolism are goldenseal and St. John's wort. The drugs that are most affected and contraindicated due to concomitant use with St John's wort are metabolized via the CYP3A4 and P-glycoprotein pathways [31]. Specifically, St. John's wort may reduce the effectiveness of cyclosporine (Sandimmune), tacrolimus (Prograf), warfarin (Coumadin), protease inhibitors, irinotecan (Camptosar), theophylline, digoxin, venlafaxine, and oral contraceptives; one should avoid combining St. John's wort with over-the-counter and prescription medications [31].

## 7. Clinical Diagnosis and Management

### 7.1. Clinical Diagnosis

Due to HDS-induced hepatotoxicity, a wide range of clinical presentations and a lack of objective diagnostic tests make it difficult for providers to establish an early diagnosis and begin early management. Patients will present with scleral icterus, abdominal pain/discomfort, nausea/vomiting, pruritus, or choluria [10]. DILI is a diagnosis of exclusion. To make a final diagnosis of DILI, a provider must obtain a detailed history and exhaust alternative diagnoses with a comprehensive work-up to distinguish competing etiologies [10].

### 7.1.1. Classifications

The Roussel Uclaf Causality Assessment Method (RUCAM), introduced in 1993, was developed to quantify the strength of association between liver injury and the medication implicated as causing the injury [32]. When it was validated, the scale demonstrated 86% sensitivity; 89% specificity; and positive and negative predictive values of 93% and 78%, respectively [32]. The scale results were reproduced by four experts after a positive rechallenge by applying RUCAM to 50 suspected DILI cases. During the experts' reproduced assessment, they found an accuracy of 99%, 74%, and 37%, respectively, which indicated a major discrepancy between expert raters [32–34]. Flaws in the scale are due to the need to assess verifiable case information, long follow-up data, and the inability to discriminate between contaminant drugs [35].

RUCAM was later modified by Maria and Victorino. This modification, the M&V scale, incorporates extrahepatic disease manifestations [36]. The overall score corresponds to five probability degrees: definite, probable, possible, unlikely, and excluded. The M&V scale was validated using real and fictitious cases and was compared with the classifications provided by three external experts. The comparison showed 84% agreement between the scale and expert opinions [36]. However, the scale is limited by the number of identified cases and unknown latency periods [36–38].

The Revised Electronic Causality Assessment Method (RECAM) is a superior update to RUCAM. It underwent 12 versions based on iterative testing, and the final scoring between RECAM and RUCAM was conducted on 98 DILIN and 96 Spanish DILI cases. RECAM had better overall agreement with expert opinion and better discriminate diagnostic categories [36].

### 7.1.2. Hy's Law

Hy's law is based on the observational work of Hy Zimmerman [37]. Zimmerman's observations showed that pure hepatocellular injury is an indicator of potential DILI. According to this observation, drug-induced jaundice caused by a drug or chemical that induces hepatocellular injury leads to death or liver transplantation in >10% of cases [37].

The FDA includes three major components of DILI in their guidance to pharmaceutical companies for the pre-marketing phase of drug safety evaluations [38]:

1. The drug causes hepatocellular injury, demonstrated by a higher incidence of the upper limit of normal ALT or AST that is three-fold greater than the (non-hepatotoxic) control drug or placebo.

2. Among trial subjects with AT elevations, one or more also show TBL serum elevations >2 × ULN without initial findings of cholestasis (elevated serum ALP).
3. No other explanation can be found for the increased AT and TBL, such as viral hepatitis A, B, or C; preexisting or acute liver disease; or another drug capable of causing the injury.

In a large population, two positive components are highly predictive that the drug has the potential to cause severe DILI [38].

The FDA also follows a standard process to evaluate liver safety in large clinical trials database using eDISH plot "evaluation of drug induced serious hepatotoxicity" [20]. eDISH provides a graphical representation broken down into four quadrants. The Temple Corollary quadrant is defined as a rise in serum ALT >3 ULN but not a concurrent rise in serum TBIL >2 × ULN [20]. This region suggests an increased risk of severe DILI.

*7.2. Management*

No data recommend a specific liver biochemistry monitoring plan when a potential hepatotoxic agent is prescribed in individuals with known CLD. The gold-standard treatment for DILI is to withdraw the offending medication. However, in severe cases, liver transplant will be indicated. Currently, there is no approved antidote for ALF resulting from idiosyncratic DILI. For these reasons, liver tests should be monitored in patients who are taking HDSs [31].

## 8. Drug-Induced Hepatotoxicity and Obesity

The increased incidence of drug induced hepatotoxicity in patients with obesity has emerged in recent clinical trials. Obesity is often associated with heart disease, T2D, hyperlipemia, non-alcohol fatty liver disease (NAFLD), and osteoarthritis. One of the hallmark features of NAFLD is hepatic steatosis [39]. Some drugs may worsen obesity-related NAFLD because of the increase in lipid deposition and necroinflammation [20]. Simultaneously, drug-induced liver toxicity may worsen due to a more hostile metabolic environment because of the increased lipid deposition and necroinflammation. This harsh metabolic environment is theorized to increase CYP2E1 expression/activity and reduce MRC activity, thus leading to an increased inflammatory state resulting in DILI [40].

## 9. Common Hepatotoxic Agents on the HDS Market

*9.1. Garcinia Cambogia*

Garcinia Cambogia (GC), also known as garcinia gummi-gutta or Malabar tamarind, is a fruit with origins in southeastern Asia. The rind is used to preserve and flavor food, and GC extract has been used more recently for weight loss, with mixed evidence [39]. The active component of GC, hydroxycitric acid (HCA), is associated with appetite suppression due to its inhibition of the ATP citrate lyase and decreased fatty acid synthesis. GC is commonly an ingredient in over-the-counter multi-ingredient dietary supplements (MIDS), such as Hydroxycut and Vi-Trim, which has made it difficult to assess its toxicity alone. However, studies in animal models indicate that there are no major toxicities aside from testicular toxicity at high doses, and rare cases of serotonin syndrome, rhabdomyolysis, and hepatic toxicity have been reported in humans [19,41].

Besides its involvement in decreased fatty acid synthesis, HCA has further downstream effects, such as increased lipid oxidation, and it may also suppress appetite via decreased serotonin reuptake [42]. A study of mouse 3T3-L1 preadipocytes treated with GC extract (59.55% HCA) found the suppression of MDI-induced adipogenesis and mitotic clonal expansion via the modulation of p90RSK and Stat3 [43]. Several clinical trials with GC alone or combined with other supplements in obese or overweight subjects reported decreases in triglycerides, cholesterol, body weight, body fat, and/or in the visceral adiposity index [44–50]. However, a review of randomized, double-blind, placebo-controlled trials evaluating GC alone found conflicting results regarding weight loss and identified several weaknesses in the study designs [42]. An earlier meta-analysis of randomized,

double-blind, placebo-controlled trials evaluating HCA alone found a borderline statistically significant decrease in body weight in HCA groups compared to the placebo groups, but the effect size was small and had uncertain clinical relevance [51]. Thus, there is a lack of robust evidence to support the use of GC as an effective weight loss supplement.

GC has been implicated in several reports of liver injury; the first case of fulminant hepatic failure from "purified" GC extract was published in 2016 [24]. Vuppalanchi et al. identified five cases of liver injury from GC alone that were reported in DILIN from 2004–2018. The liver injury pattern was hepatocellular in four cases and mixed in one case. All the patients were hospitalized, and one required a liver transplant. The number of days from the start of drug consumption to the onset of liver injury ranged from 35 to 139. When combined with green tea or ashwagandha, there were 22 cases of liver injury associated with GC. There were no significant differences in the clinical characteristics or outcomes between the cases related to GC alone and the cases of GC with green tea; both groups presented with acute hepatitis [52]. Since 2019, there have been at least five case reports of liver injury associated with weight-loss supplements containing GC [53–57]. GC has a DILIN category B likelihood score (likely rare cause of clinically apparent liver injury), with a frequency of <1:10,000 for hepatic adverse reactions. The exact mechanism of hepatotoxicity is unknown; it appears to be idiosyncratic and generally resolves 1–3 months after discontinuation [41]. HCA increases hepatic collagen accumulation and lipid peroxidation, and it is involved in inflammatory responses and oxidative stress, which may lead to hepatocellular injury [58].

*9.2. Saw Palmetto*

*Serenoa repens*, commonly known as saw palmetto (SP), was the 11th highest selling herbal supplement in the US in 2020 [59]. This supplement is derived from the berries of a palm-like plant that is native to Florida, and the southeastern US SP supplements are available in a variety of forms, including dried berries, powdered capsules, tablets, liquid tinctures, and liposterolic extracts. In 2020, global SP supplement sales totaled over USD 117 million.

SP supplements are mainly used to improve urinary symptoms related to benign prostatic hyperplasia (BPH), although there are not sufficient randomized controlled studies in human subjects to conclude its claim [58]. The mechanism of action for SP is not well understood. Many believe that it works through inhibiting 5α-reductases and that it may also have other anti-androgenic and anti-inflammatory effects [60]. Studies have demonstrated that SP has α1-adrenoreceptor inhibitory effects [61]. Studies on rat models have shown that SP works on the α1-adenoreceptors and muscarinic receptors in the lower urinary tract to relieve dysuria symptoms [62]. However, multiple double-blind RCTs have shown no significant improvements in urinary symptoms using SP versus a placebo [63,64]. SP supplements are also used to treat alopecia and hair loss. The mechanism behind this is thought to be related to SP's antiandrogenic properties. Several RCTs have shown reductions in hair loss and improvements in hair quality using SP supplements [57,63,64]. SP supplements have also been used to enhance sperm production, breast size, and libido and may also function as a mild diuretic [65]. However, almost all current formulations containing SP also contain other vitamins, minerals, or chemical additives, making it challenging to discern the exact extent of SP's contribution to observed clinical findings.

The most common side effects of SP supplements include headache, fatigue, abdominal pain, nausea, vomiting, diarrhea, and decreased libido [66]. Studies have shown that SP supplements do not affect the hepatic clearance of co-administered medications and have not been linked to liver enzyme disturbances with long-term use [67,68]. Three cases of acute liver injury that can be attributed to SP have been reported [68–70]. In each case, liver function returned to baseline upon discontinuing the supplement. SP has been linked to acute pancreatitis in two separate case reports [71,72].

### 9.3. Ashwagandha

Ashwagandha is an Ayurvedic herb derived from the roots of Withania somnifera, which is an evergreen shrub found in India and Southeast Asia. For centuries, it has been used in Ayurvedic medicine, but its use as an herbal supplement in the US has increased in recent years [73]. The mainstream channel sales of ashwagandha totaled nearly USD 11 million in 2019, a 45.2% increase from 2018, and it was the fifth highest selling herbal supplement in the US Natural Channel [74]. Ashwagandha's purported benefits include increased energy, reduced fatigue, and anti-inflammatory effects, and it is used for conditions such as stress, skin diseases, diabetes, arthritis, and epilepsy [73].

Clinical trials have demonstrated various effects of ashwagandha, such as improved sleep quality [75–77]; improved cognitive performance [78], quality of life, and mental alertness [76] in elderly subjects; decreased HAM-A scores in subjects with anxiety [79,80] and high stress [81]; increased $VO_{2max}$ in healthy adults and athletes [82,83]; decreased Perceived Stress Scale scores and serum cortisol in stressed adults [78,82,84]; and improved depression and anxiety scores [85] and Perceived Stress Scale scores [86] in individuals with schizophrenia or schizoaffective disorder. Due to its stress-reducing effects, ashwagandha is considered an adaptogen, a substance that helps the body respond and adapt to stress and environmental changes [84]. However, external validity, experimenter bias, publication bias, lack of follow of follow up and poor study designs limit reported benefits.

There are few adverse effects associated with ashwagandha, although several cases of mostly mild-to-moderate liver injury in patients taking commercial herbal products containing ashwagandha have been reported in recent years [73]. A case series reviewed five cases of cholestatic or mixed liver injury from 2016 to 2018 that were accompanied by jaundice, pruritus, and hyperbilirubinemia [87]. Symptoms occurred 2–12 weeks after starting the supplement, and liver tests normalized within 1–5 months of stopping use. The one biopsy that was performed showed acute cholestatic hepatitis, and liver failure did not occur in any of the cases. Since then, several more reports of ashwagandha-related liver injury have emerged [88,89], with one case of ALF requiring transplantation [90]. The patient in this case however was also taking an unknown dose of progesterone; thus, it is unclear whether this drug also played a role in the development of ALF. This case and one of the other cases showed a hepatocellular injury pattern that contrasted the previously described cholestatic or mixed patterns. Ashwagandha has a DILIN category C likelihood score (probable cause of clinically apparent liver injury). Its mechanism of hepatotoxicity is unclear, although the metabolite withanone appears to form adducts with several nucleosides, DNA, and amines and thus may cause DNA damage in low-GSH conditions [91]. Research in rats suggests that Withania somnifera has a hepatoprotective effect that is caused by decreasing ALT, AST, ALP, MDA, and bilirubin and increasing the antioxidant capacity [92].

### 9.4. Green Tea

Green tea is produced from the leaves of the *Camellia sinensis* plant, which is found primarily in South and Southeast Asia. It is unfermented, which preserves its antioxidant polyphenolic catechols. Tea catechins are the bioactive phytochemicals responsible for the antioxidative properties of green tea, with (−)-epigallocatechin-3-gallate (EGCG) being the most abundant, active, and hepatotoxic species [93]. GTE contains a concentrated amount of these substances. Increased production and a rising demand have increased GTE use in recent years [7]. Interest in the antiviral activity of EGCG also increased during the pandemic, as people considered its use against SARS-CoV-2 [94]. In 2020, consumers in the US spent over USD 31.4 million on green tea HDSs [7].

The proposed benefits of green tea stem from its antioxidant activity; low incidence of high-oxidative-stress pathologies such as cardiovascular disease and cancer; and its potential to improve the metabolism of glucose, lipids, and uric acid [95]. Green tea has been marketed for a variety of conditions, from cancer prevention to weight loss, with varying scientific support [96]. Green tea accounts for more than 20% of global tea production,

and GTE is frequently used in HDSs [97]. GTE is the main ingredient in heavily marketed over-the-counter weight-loss supplements such as Slimquick, Hydroxycut, Herbalife, the Right Approach, and Ripped Fuel. GTE formulations and concentrations vary amongst different HDS products. While consuming green tea on its own is relatively innocuous, it is the consumption of GTE that is a cause for concern.

Given its increasing popularity, it is important to define the safety profile of GTE. although rare, GTE has been linked to more than 100 reports of characteristic acute hepatitis-like illness, which presents with acute hepatocellular injury and jaundice [98]. It has a DILIN category A likelihood score, indicating that it is a well-established cause of clinically apparent acute liver injury, usually arising within 1 to 6 months of starting the product and resolving within 1 to 2 months after stopping use [98]. Recurrence is possible upon re-exposure, and the aminotransferases, alkaline phosphatase, and bilirubin levels react accordingly. The use of the weight loss supplement Slimquick has led to fulminant liver failure and orthotopic liver transplantation in some cases, while fatal instances have been reported in up to 10% of cases [10,99]. However, there is a dearth of research regarding GTE's effects in different populations. Considering the wide range of injuries, it is necessary to explore the circumstances that lead to these adverse effects. Specific genes in mice were found to be associated with susceptibility to EGCG toxicity, and analogous human genetic variants may show the same relationship [1].

GTE's potential for therapeutic benefits must be weighed with its risk of hepatotoxicity. This balance is illustrated in the treatment of cirrhosis caused by chronic hepatitis C virus (HCV) infection. In one prospective, randomized, double-blind study involving 338 Japanese patients, cyanidanol appears to have increased host anti-HBV responses and improved host clearing of infected hepatocytes [100]. While patients may benefit from the antiangiogenic, antioxidant, and antifibrotic properties of EGCG, the reported dose-dependent potential for hepatotoxicity could be further augmented in cases in which cirrhosis is present [101]. However, further clinical studies must be investigated.

Recently, with a rise in obesity, GTE consumption has been investigated in comparison to fasting and fad states. The study showed that EGCG caused dose-dependent hepato-toxicity in mice under dietary restriction, suggesting a combination of effects of dietary restriction and EGCG. It was hypothesized that the disturbance of lipid metabolic pathways from dieting and the disruption of the Lands' cycle led to liver injury. However, these studies were limited to animal trials [102]. There are few clinical studies in human subjects, and further investigation is necessary.

### 9.5. Aloe Vera

Aloe vera is a cactus-like plant in the lily family that is used both topically and orally. Aloe vera products come from the leaf and contain a wide range of substances such as anthraquinones; vitamins A, C, and E; enzymes; salicylic acid; and hormones. Topical creams that include aloe are used for cosmetic purposes and wound healing, and its extracts are used for food flavoring and in dietary supplements. Mainstream channel aloe vera sales totaled USD 21 million in 2019, and it became the sixth highest selling herbal supplement in the US Natural Channels [74]. Aloe vera is purported to aid in weight loss and in conditions such as arthritis, asthma, chronic fatigue, dyspepsia, constipation, and skin ailments. Furthermore, the antiproliferative, anti-inflammatory, and hepatoprotective effects of aloe vera have been indicated in several animal and in vitro studies [103]. Hepatoprotective effects have been demonstrated in mice with acetaminophen-induced hepatitis [104] and in rats with non-alcoholic steatohepatitis [105], ischemia-reperfusion injury [106], alcohol-induced hepatic damage [107], and oxidative stress-induced damage [108].

There are no major adverse effects associated with topical aloe vera aside from rare hypersensitivity reactions, skin rash, and allergic dermatitis. However, oral preparations have been associated with at least a dozen cases of liver injury since 2005. The pattern of liver injury was primarily hepatocellular and had a presentation similar to acute viral hepatitis [103]. A review of nine cases of aloe vera-induced liver injury found a latency that

ranged from 2 weeks to 6 months, and the majority had a probable RUCAM score [109]. The overall pattern in the six liver biopsies that were performed indicated portal and acinar inflammatory cell infiltration with bridging necrosis and bile stasis [109]. A systematic review of herb-induced liver injury identified 22 cases associated with aloe vera, in which 4.1% were treated with liver transplantation, 4.7% were treated with chronification, and 9.5% resulted in death. However, liver transplantation and death occurred in patients consuming multiple herbs. The injury was generally self-limiting and resolved after stopping aloe vera consumption, with n liver tests typically normalizing within 50 days [110].

The hepatic injury caused by aloe vera appears to be idiosyncratic; none of the individual components of the leaf extracts are known to be hepatotoxic [103]. In one study, healthy volunteers consumed 2 oz of aloe vera gel twice daily for 60 days, and no changes in the biochemical indices of liver function, ultrasound markers of hepatic blood flow, or liver tissue elasticity were found [111]. Liver injury from aloe vera may be related to hypersensitivity, as eosinophilic infiltrates were noted in several biopsies [109].

### 9.6. Germander

Teucrium chamaedrys, an herb commonly called germander, has been used for many years to treat gastrointestinal illness, fevers, gout, and other forms of inflammation. More recently, it was marketed to address weight loss and elevated cholesterol; however, studies on these effects are lacking [19]. Germander has been identified as a likely cause of acute liver injury and is associated with several cases of ALF necessitating transplant [19]. Most cases of acute liver injury resolve once germander supplementation is stopped, but some patients have died from ALF caused by germander [19]. Latency of acute liver injury for germander averages around 9 weeks but can take as little as 2 weeks or as long as 18 weeks. Patients with germander-induced liver injury present with many similarities to a case of acute viral hepatitis including nausea, jaundice, and a hepatocellular pattern of liver enzyme elevation. Liver biopsies showing centrilobular necrosis were found in patients suffering from this acute germander liver injury [19]. Germander has also been associated with a presentation similar to chronic hepatitis when taken over the course of months. The symptoms of this more chronic presentation may include arthralgias and fevers, while serum tests may show autoantibodies. Fibrosis was seen on biopsies in these cases [19]. Germander contains neoclerodane diterpenoids, whose metabolites are the probable cause of its hepatotoxicity. Neoclerodane diterpenoids use CYP450 for metabolism, creating metabolites that deplete glutathione, attach to proteins, and interfere with many other cellular activities. Similar to acetaminophen, this substance harms the liver with a mechanism that depletes glutathione. Hepatotoxic effects recur in many cases when patients are again exposed to germander [19].

### 9.7. Vitamin A

Vitamin A is a fat-soluble molecule that is a dietary requirement for normal functioning of the eyes, skin, bones, and immune system. It has a recommended daily allowance (RDA) of between 700 and 900 μg per day in adults and can have toxic effects when people take amounts that are significantly higher than their daily allowance. Vitamin A has a DILIN likelihood score of A, the highest score on this scale, indicating that it is a well-known cause of liver injury [19]. It is often taken in the form of a pill, tablet, or a solution but may also be found in many vitamin and mineral combinations and herbal supplements [19]. Vitamin A is also indicated and often prescribed to treat measles, xeropthalmia, and malnutrition, and is sometimes given to pregnant women to make sure they are not deficient [112]. Hepatotoxicity arises with the surreptitious use of vitamin A, not under a provider's guidance. Signs of toxicity include hepatosplenomegaly, jaundice, portal hypertension, and liver injury. Toxicity can be acute or chronic in nature depending on how the vitamin A is ingested. Acute cases can happen within a matter of days of large-dose ingestion, while chronic cases may take many months to develop. Treatment involves discontinuing supplementation and starting a low vitamin A diet [112].

*9.8. Black Cohosh*

Black cohosh (BC) is often used to help mitigate menopause symptoms and has been shown to be effective for this in some clinical trials [19]. Products labeled as BC have been associated with liver injury, ranging from mild transaminases to cases of autoimmune hepatitis, ALF, and death [19]. The mechanism of action is not fully understood, but it is theorized to be an immunologic reaction. BC has well-established causes of clinically apparent liver injury, giving it a likelihood score of A. It is difficult to isolate a cause in these cases because many HDSs contain adulterants or mislabeled products. Most cases improve with discontinuation of the HDS, but steroids or immunosuppression may be needed in cases with long lasting autoimmune symptoms [19].

*9.9. Turmeric*

Turmeric comes from the roots of Curcuma longa, a ginger family plant native to India and grown in southern Asia and central America [19]. In addition to its use as a dye and spice for curry, turmeric and its purified extract curcumin are used in Ayurvedic medicine to treat gastrointestinal issues and upper respiratory infections due to their potential anti-inflammatory, antineoplastic, and antioxidant effects [19]. Specifically, curcumin decreases proinflammatory cytokines; induces heme oxygenase-1 expression and reduced glutathione synthesis; and regulates numerous antioxidant signaling pathways [113].

Prior to the recent publication of case reports of turmeric-associated liver injury, turmeric had been known to temporarily increase serum enzymes without symptoms or clinically apparent acute liver injury [19]. Curcumin has been investigated as a hepato-protective agent and potential treatment for oxidative associated liver diseases, with two clinical studies showing efficacy in NAFLD but lacking histological endpoints [113–115]. Another RCT found that lifestyle modifications plus curcumin significantly reduced hepatic fibrosis and steatosis scores, but these changes were not significantly different from the placebo group [116].

The emergence of cases of turmeric-associated liver injury may be attributed to the introduction of high bioavailability turmeric via piperine (from black pepper) or nanoparticle delivery [19]. A systematic review and meta-analysis published in 2020 described seven cases of acute hepatitis in Tuscany, Italy, from 2018–2019 associated with dietary supplements containing Curcuma longa, and it identified 23 other cases from 13 case reports in multiple countries [117]. All cases in Tuscany involved high bioavailability formulations, especially with piperine, which increases curcumin absorption by 200% [117]. Six of the seven patients were female, and the age range was 45–68 years. ALT and AST were >1000 in six out of seven and five out of seven cases, respectively; the daily dose was 1000 mg or greater in four out of seven cases. Positive dechallenge occurred in five out of seven cases, and six of the seven cases had RUCAM and WHO-UMC categories of probable or possible (1 unlikely/not assessable). All but one of the thirteen studies in the systematic review mentioned the concomitant use of other medications/products, and positive dechallenge was observed in 17 of 23 cases.

The pattern of liver injury is hepatocellular and presents similarly to autoimmune hepatitis clinically and histologically [19]. Turmeric has a DILIN category B likelihood score (likely rare cause of clinically apparent liver injury) [19]. The mechanism of liver toxicity is not well understood and may be idiosyncratic; it is likely not an immunological reaction [117]. Interaction with other drugs may also play a role in turmeric-associated liver injury [117].

Summary of 9 common hepatotoxic HDSs on the market with their theoretical benefits (Figure 1).

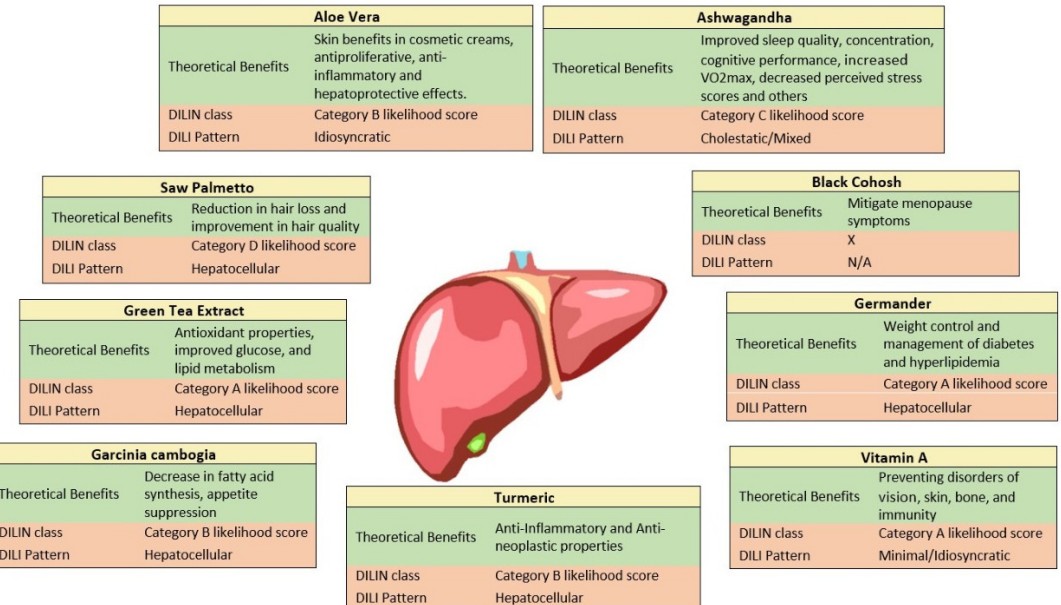

**Figure 1.** Summary of 9 common hepatotoxic HDSs on the market with their theoretical benefits; DILIN class, i.e., likelihood of causing clinically significant liver injury; and the pattern of resultant liver injury. Likelihood score obtained from the DILI Network LiverTox database [19,39,74,100,103]. It is a 5-point scale (A to E) that estimates whether a medication is a cause of liver injury: A = well-known cause; B = highly likely cause; C = probable cause; D = possible cause; E = unlikely cause; E* = suspected but unproven cause; X = unknown.

## 10. Limitations

A critical review of individual studies is important because of increased theoretical claims and side effects. Table 3 summarizes the limitations of a few studies discussed in our review. Many studies presented in our paper have limited adequately powered clinical studies in human subjects, external validity, experimenter bias, publication bias, lack of follow, and short clinical trials that create a crucial need for more robust reporting processes for the adverse hepatotoxic effects caused by HDSs. There is a need for critical review on clinical trial designs and further investigating studies regarding the likelihood of hepatotoxicity of certain FDA-approved HDSs. We have presented a brief overview of multiple case reports that have reported herbal induced liver injury. However, case reports carry major limitations such as establishing a cause–effect relationship, danger of over-interpretation, and publication bias [118].

**Table 3.** List of individual studies and their limitations.

| Study | Claim | Major Limitation |
|---|---|---|
| Feng et al., 2021 [3] | Zinc sulfate shortens the duration of olfactory dysfunction in patients with COVID | Low quality of selected trials; only RCTs were included and had a wide variation with the targeted population, mode of delivering HDS, outcomes, and follow-up periods |
| Lunsford et al., 2016 [24] | Fulminant hepatic failure caused by Garcinia Cambogia 5:1 extract | No independent laboratory evaluation of supplement to identify contaminants/verify composition |
| Han et al., 2021 [43] | Garcinia cambogia extract suppresses adipogenesis in mouse 3T3-L1 preadipocytes via modulation of p90RSK and Stat3 | In vitro study |
| Maia-Landim et al., 2018 [44] | Garcinia cambogia and Glucomannan reduce weight and improve lipid and glucose blood profiles in overweight/obese individuals | Not randomized; combination therapy with GC and Glucomannan; included patients with dyslipidemias, hypertension and type 2 diabetes receiving treatment for 2 months to 4 years before study onset |

**Table 3.** *Cont.*

| Study | Claim | Major Limitation |
|---|---|---|
| Chong et al., 2014 [45] | IQP-GC-101 reduces body weight and body fat mass in overweight Caucasian adults | IQP-GC-101 includes 4 different extracts including GC; did not exclude subjects with regular caffeine intake; no strict control of daily caloric intake |
| Vasques et al., 2014 [46] | Short-term Garcinia cambogia treatment significantly reduces triglyceride levels in obese women without affecting anthropometric or calorimetric parameters | Small sample size; did not use intention-to-treat methods; did not describe dropouts; did not include *p* values for all comparisons |
| Vasques et al., 2008 [47] | Garcinia cambogia plus, *Amorphophallus konjac* treatment significantly reduces total cholesterol and LDL-c levels in patients with obesity compared to placebo without affecting anthropometric or calorimetric parameters | Combination therapy with GC and *Amorphophallus konjac is* relatively small sample size. |
| Toromanyan et al., 2007 [48] | Slim339 significantly reduces body weight in overweight and obese individuals | Slim339 contains 5 extracts and calcium pantothenate; relatively small sample size and short treatment period |
| Opala et al., 2006 [49] | Botanical extract-based weight loss formula produces significant change in the Body Composition Improvement Index and decrease in body fat in healthy, overweight subjects | Two different tablets with different combinations of herbal extracts (GC in one tablet); % body fat estimated with 4-skinfold method; more female than male subjects (77 vs. 21); smoking not in exclusion criteria (but monitored) |
| Tallei et al., 2021 [94] | Green tea extract may serve as a potential treatment against COVID-19 | In vivo studies have not been studied. |
| Peluso et al., 2017 [95] | The consumption of green tea can modulate the antioxidant capacity of individuals | No convincing evidence from long-term intervention studies in humans |
| Jurgens et al., 2012 [96] | Green tea preparations induce weight loss in overweight or obese adults | Some studies had incomplete reporting and short study period (12 weeks) |
| Whitsett et al., 2014 [99] | Case report of fulminant liver failure and orthotopic liver transplantation in a patient after Slimquick use | Danger of over-interpretation and publication bias. Not a powered clinical trial. |
| Isbrucker et al., 2006 [102] | EGCG caused dose-dependent hepatotoxicity in mice under dietary restriction | Studies were limited to animal trials |
| Peng et al., 2016 [119] | Eight randomized studies with 800 patients and showed positive effects on the levels of ALT, AST, TC and TG, LDL. | Limited to the duration of follow up and small sample size to confirm the efficacy and safety of the study |
| Deshpande et al., 2020 [75] | Ashwagandha Improved sleep quality in meta-analysis | Short trial period to establish long term effects, limited trials presented with a potential risk of publication bias |
| Dimpfel et al., 2020 [78] | Ashwagandha Improved cognitive and concentration performance | Insufficient details regarding preparation of the extract, treatment administration, and randomization procedures |
| Hoogenboom et al., 2020 [111] | hepatic injury caused by aloe vera appears to be idiosyncratic; no changes in the biochemical indices of liver function | Not all participants completed the trial, few missed doses for some participants, trial lasted 60 days, low number of participants |

## 11. Conclusions

Considering increased HDS popularity and sales, it is crucial for clinicians to challenge their use. With many hypotheses, conflicting data, and increased speculation, it is imperative for researchers to fill in the gaps of our current knowledge of HDS hepatotoxicity.

Although there is potential for hepatotoxicity, some studies have suggested different outcomes. Certain herbal supplements have shown promising effects, such as Salvia Miltorrhiza Bunge. A meta-analysis of eight randomized studies with 800 patients showed positive effects on the levels of ALT, AST, TC and TG, LDL, and the liver/spleen-computed tomography ratio in patients with NAFLD [116,118,119]. However, the study was limited to the duration of follow up and would require a larger sample of randomized clinical trials to confirm the efficacy and safety of the study.

Additional research on biomarkers can help validate new sensitive and diagnostic testing for early cases. A recent study showed that pyrrolizidine alkaloids (PAs), which are natural toxins synthesized from different plant species, are considered toxic but are widely found in food and herbal products. PAs were found in herbal infusions containing comfrey, green tea, and honey [120]. More recent studies have found three potential biomarkers (hsa-miR-148a-3p, hsa-miR-362-5p, and hsa-miR-194-5p) in patients with PA-induced hepatic sinusoidal obstruction syndrome (HSOS) [121]. Further studies in this area are warranted.

Nonetheless, there is a need for a more complete understanding of the hepatoxic effects of HDSs in the general population and specifically in patients with CLD. The impact of the hepatotoxic effects caused by HDSs continues to be unclear because many forms of liver disease can be mimicked by drug-induced liver injury.

**Author Contributions:** Conceptualization, A.K., D.H.-D. and K.C.; Methodology, A.K. and D.H.-D.; Resources, A.K. and K.C.; Writing—original draft preparation, A.K., K.C., H.R., N.S.P., J.M., M.W., P.Z. and R.E.; Writing—review and editing, A.K., K.C. and D.H.-D.; Supervision, D.H.-D. All authors have read and agreed to the published version of the manuscript.

**Funding:** This research received no external funding.

**Conflicts of Interest:** The authors declare no conflict of interest.

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
