# Peer review of "A Comprehensive Review on the Use of Herbal Dietary Supplements in the USA, Reasons for Their Use, and Review of Potential Hepatotoxicity"

_livers, doi:10.3390/livers2030011_

Round 1

Reviewer 1 Report

[*] What is the main question addressed by the research?

This is a review on the use of herbal supplements in the USA. It is a very interesting topic, and the review is well written. It does attract attention to this major health problem and situates the reader on what is going on right now over this theme.

[*] Is it relevant and interesting?

Surely, it is very relevant given the increase in use of herbal supplements.

[*] How original is the topic?

It is not very original. Many reviews on the subject have been reported recently, even a systematic review comprised of many cases of HILI: "https://www.wjgnet.com/2307-8960/full/v9/i20/5490.htm"

[*] What does it add to the subject area compared with other published

material?

Not much. Although it is well written, it is able to gather relevant information and summarize it well.

[*] Is the paper well written? Yes

[*] Is the text clear and easy to read? Yes

[*] Are the conclusions consistent with the evidence and arguments

presented? Yes

[*] Do they address the main question posed?

Yes

Author Response

Thank you so much for reviewing our paper. 

Reviewer 2 Report

This review article highlighted an important issue of hepatotoxicity by herbal and dietary supplements and stressed for further investigations and relevant policy making. Overall, the study is well-reviewed and drafted. I have one suggestion to further improve the readability of this manuscript:

1. A table may be added highlighting key findings and possible explanations, if any, of all the case reports discussed in the manuscript.

Author Response

Thank you so much for reviewing our paper.

We have included a Table 1 and Table 2 to help understand the enzymes involved in the livers metabolism. 

Reviewer 3 Report

--Summary figures that show the phases of drug metabolism and disposition and the likely immune-mediated pathogenesis of most HDS- or drug-induced liver injury would improve readability and impact.

--A more critical assessment of the publications cited regarding purported benefits of HDS would improve the MS.  There are many, many claims of benefit, but very few, if any,  that are supported by placebo-controlled and adequately powered clinical studies in human subjects.

--The importance of obesity and/or fasting vs fed state for HDS toxicities, such as from green tea extracts, needs to be discussed.

--the increasing incidence of HILI ascribed to turmeric, especially when taken with black pepper, which increases its absorption, deserves mention and discussion.

--There should be mention of the Delphic approach to causality assessment [Rochon et al, Hepatology] as superior to RUCAM.  There should be mention of the newer RECAM method [Hayashi et al, Hepatology, 2022].

--The important review of Seeff et al, Gastroenterology, deserves citation and discussion.

--It would be better if the authors were able to cite more recent data on HDS use than 2012 [ref 1] and 2018 [ref 2].

--Although generally clear, there are several errors and ambiguities in need of correction:  line 117--for 'transaminitis', read 'transaminases;'

--l 119--the principal Phase 1 reaction is hydroxylation;

--l 122--contributing to metallothionein?? How?? Why??

--l 179--deleted second 'serum;'

--l 243, l 387--pruritus is correct;

--l 269--Define Hy's Law; mention Temple's corollary;

--Several references are formatted incorrectly;

--l 689--should read androgenic'

--l 691--should read comparative.

Author Response

First, I would like to say thank you so much for reviewing our paper. 

To review each point

--Summary figures that show the phases of drug metabolism and disposition and the likely immune-mediated pathogenesis of most HDS- or drug-induced liver injury would improve readability and impact.

- We have included a summary table 1 and table 2 to help understand the enzymes involved in both Phase 1 and Phase 2.  I have also included a paragraph to discuss the immune mediated pathogenesis at end of section 3. 

--A more critical assessment of the publications cited regarding purported benefits of HDS would improve the MS.  There are many, many claims of benefit, but very few, if any,  that are supported by placebo-controlled and adequately powered clinical studies in human subjects.

- we have made sure to really emphasize in our discussion and through out the paper in how there are many beneficial claims but unfortunately there are limited studies in human subjects. We wanted that point to be one of the major take aways from our paper. 

--The importance of obesity and/or fasting vs fed state for HDS toxicities, such as from green tea extracts, needs to be discussed.

We included a new section (Drug Induced Hepatotoxicity in Obesity). We read a very detailed and thought provoking text book by Dr. Ramachandran and Dr. Jaeschke, Drug-Induced Liver Injury that really helped explain this concept. in addition, we also included in our paper a very intriguing study that discussed how EGCG caused dose-dependent hepatotoxicity on mice under dietary restriction. Included in at the end of Green Tea Section. Thank you for this comment. 

--the increasing incidence of HILI ascribed to turmeric, especially when taken with black pepper, which increases its absorption, deserves mention and discussion.

- We have included a whole section on Tumeric and discussed a clinical study showing black pepper increasing  Curcumin absorption. We do agree that this deserves attention. 

--There should be mention of the Delphic approach to causality assessment [Rochon et al, Hepatology] as superior to RUCAM.  There should be mention of the newer RECAM method [Hayashi et al, Hepatology, 2022].

- Under the section classification, we have included a whole paragraph on RECAM

--The important review of Seeff et al, Gastroenterology, deserves citation and discussion.

- we read, L. B. Seeff Seeff et al, Gastroenterology, which was a very well written paper and included a prospective, randomized, double-blind study involving 338 Japanese patients showing cyanidanol appears to have increased host anti-HBV responses and to have improved host clearing of infected hepatocytes. however, further clinical studies and trials are imperative. We do agree that the article deserves citation/discussion.

--It would be better if the authors were able to cite more recent data on HDS use than 2012 [ref 1] and 2018 [ref 2].

- We strongly feel that our resources are up to date in the areas we wanted to discuss. Nonetheless, we have updated some resources to make sure that our paper is a comprehensive and updated review.

--Although generally clear, there are several errors and ambiguities in need of correction:  line 117--for 'transaminitis', read 'transaminases;'

--l 119--the principal Phase 1 reaction is hydroxylation;

- Made sure to add hydroxylation in the text

--l 122--contributing to metallothionein?? How?? Why??

- This was a mistake that was deleted. There is no contribution rather MT is different pathway that is induced by gene expression. We included MT more in detail under Other pathways. 

--l 179--deleted second 'serum;'

- Deleted

--l 243, l 387--pruritus is correct;

- Fixed

--l 269--Define Hy's Law; mention Temple's corollary;

- We included temple corollary and redefined hys law by reading Dr. Ramachandran and Dr. Jaeschke, Drug-Induced Liver Injury. It is located at the end of "7.1.2. Hy’s Law". We also discussed eDISH which deserved to be discussed.

--Several references are formatted incorrectly;

--l 689--should read androgenic'

--l 691--should read comparative.

---^ for all of the following comments, We reached out to LIVERS editing service to fix all of our citation and grammatical errors. 

Once again, Thank you so much for reviewing our paper 

Round 2

Reviewer 3 Report

--Unfortunately, many, many errors of English usage, misspellings, etc, continue to occur throughout the MS. They are too numerous to list. A few examples include the following: Section 7.1--the first sentence is unclear and poorly written; the word pruritus continues, at times to be misspelled as 'pruritus';for 'discriminate', read 'discriminant.'

--Section 6.--What is the evidence that 'cytochrome P450 enzymes regulate gene expression and enzyme activities'?  Which genes and which enzymes? How and why?

--Section 7.1.2--Hy's Law does NOT predict a worst case DILI outcome.  Rather, it is a clinical observation that, for drugs and chemicals that cause an hepatocellular-type of liver injury, with the patient develops clinically apparent jaundice [usually with serum total bilirubin greater that 3 mg/dL, the lower limit of clinical detection in most patients, there is ~ 10% risk that the patient will die of the injury.

--Figure 2 now appears twice; why?

--Section 9.3--What is 'mainstream channel sales'? What is 'Natural Channel sales'?

Author Response

Hope everyone is doing well.

I am a little confused and concerned about reviewer 3 comments for our paper A Comprehensive Review on the Use of Herbal Dietary Supplements in the USA, Reasons for their Use and Review of Potential Hepatotoxicity. livers-1766472.

I read through the review and am concerned about the fairness of the review. 

Comments "Unfortunately, many, many errors of English usage," and Figure 2 now appears twice; why?". 

  • First, I would like to say, we have used MDPI editing service twice for our paper. (see attached). 
  • Second, As a reviewer myself, I do not believe the reviewer understands that the document shows both "corrected" and "incorrected" version of the paper because we were supposed to track our changes. (reason why there are 2 figure 2s)

If there continues to be errors, I will be happy to send our paper back to MDPI editing service to make changes. 

Other comments: What is the evidence that 'cytochrome P450 enzymes regulate gene expression and enzyme activities'?  Which genes and which enzymes? How and why?

  • Our paper purpose is to provide an overview use of HDS and reasons for their hepatotoxity which we have explained. Further explaining the biochemistry of molecular pathways will distract the reader from the main point of the paper. nonetheless, I am happy to explain it further.

However, I am happy to make changes to Hys law definition (as there might be some controversial definitions) and explain what 'mainstream channel sales and 'Natural Channel sales". Which I believed were common terminology. 

Looking at the reviewer prior review, We have made all the changes he requested to make our paper more impactful.